# Structural reconstruction of individual filaments in Aβ₄₂ fibril populations assembled in vitro reveal rare species that resemble ex vivo amyloid polymorphs from human brains
Liam D. Aubrey [1,3], Liisa Lutter [1,4], Kate Fennell [2], Tracey J. Purton[1], Natasha L. Ward[1], Louise C. Serpell [2] & Wei-Feng Xue [1] ✉

Structural polymorphism has been demonstrated for both in vitro and ex vivo amyloid fibrils associated with disease. The manner in which different filament structures are assembled from common building blocks remains unclear but the assembly environment is likely to be a key determinant. To address this, three-dimensional reconstruction of individual filament structures was conducted from atomic force microscopy images to map the structural polymorphism landscape of Aβ₄₂ amyloid fibril populations formed in vitro under most frequently used buffer conditions. The data shows sensitivity of Aβ₄₂ fibril polymorphism to the assembly environment in both the magnitude of heterogeneity and the types of filament species formed. However, some conserved fibril polymorphs were observed across the experimental conditions. Excitingly, by matching individual filament structures to cryo-electron microscopy derived structural data, rare species in these heterogeneous population clouds that show remarkable similarity to Aβ₄₂ amyloid polymorphs purified from human patient brains were discovered. These results link in vitro experimental approaches with structures formed in vivo, and highlight the polymorph distribution, and the type and magnitude of structural variations within these heterogeneous molecular distributions as important factors in amyloid biology.

Protein misfolding and the formation of amyloid fibril structures is associated with pathology in numerous diseases[1,2]. This includes, but is not limited to, neurodegenerative diseases such as Alzheimer's disease (AD), in which extracellular deposits known as amyloid plaques are found in brains of patients. The plaques are composed of amyloid fibrils made from cleavage products of the amyloid precursor protein (APP)[3]. Amongst these fragments, the 42 residue amyloid β 1-42 peptide (Aβ₄₂) is considered to be an important aggregation prone fragment that has been shown to produce species which are toxic to neuronal cells[4,5] and fibrils that are part of insoluble deposits in human brains[6]. Aβ₄₂ amyloid fibrils, like all amyloid fibrils, are defined by their core cross-β molecular architecture composed of β-strands that stack perpendicular to the long fibril axis to form protofilaments

up to several microns in length[7,8]. Multiple protofilaments can laterally associate to form twisted fibrils with a hydrophobic core[9–12]. Various experiments have been performed to determine the specific structural properties of Aβ fibrils. Atomic-detailed structures have been generated from solid state NMR (ssNMR) (e.g.[13–17]) and cryo-transmission electron microscopy (cryo-TEM) (e.g.[6,18–26]). Interestingly, different structural polymorphs have been observed for fibrils formed from both amyloid β 1-40 peptide (Aβ₄₀)[25,26] and Aβ₄₂[6,22]. In one of the studies of ex vivo Aβ₄₂ fibrils purified from human brains[6], multiple left-hand twisted polymorphs were observed in the same samples originating from the same patients (Fig. 1a, b). Contrastingly, in patient brain derived Aβ₄₀ fibrils from a meningeal sample[25], the most populous fibril structure was a right-hand twisted fibril

[1]School of Natural Sciences, University of Kent, Canterbury, UK. [2]Sussex Neuroscience, School of Life Sciences, University of Sussex, Brighton, UK. [3]Present address: Astbury Centre for Structural Molecular Biology, School of Molecular and Cellular Biology, Faculty of Biological Science, University of Leeds, Leeds, UK. [4]Present address: Molecular Biology Institute, UCLA, Los Angeles, CA, USA. ✉e-mail: W.F.Xue@kent.ac.uk

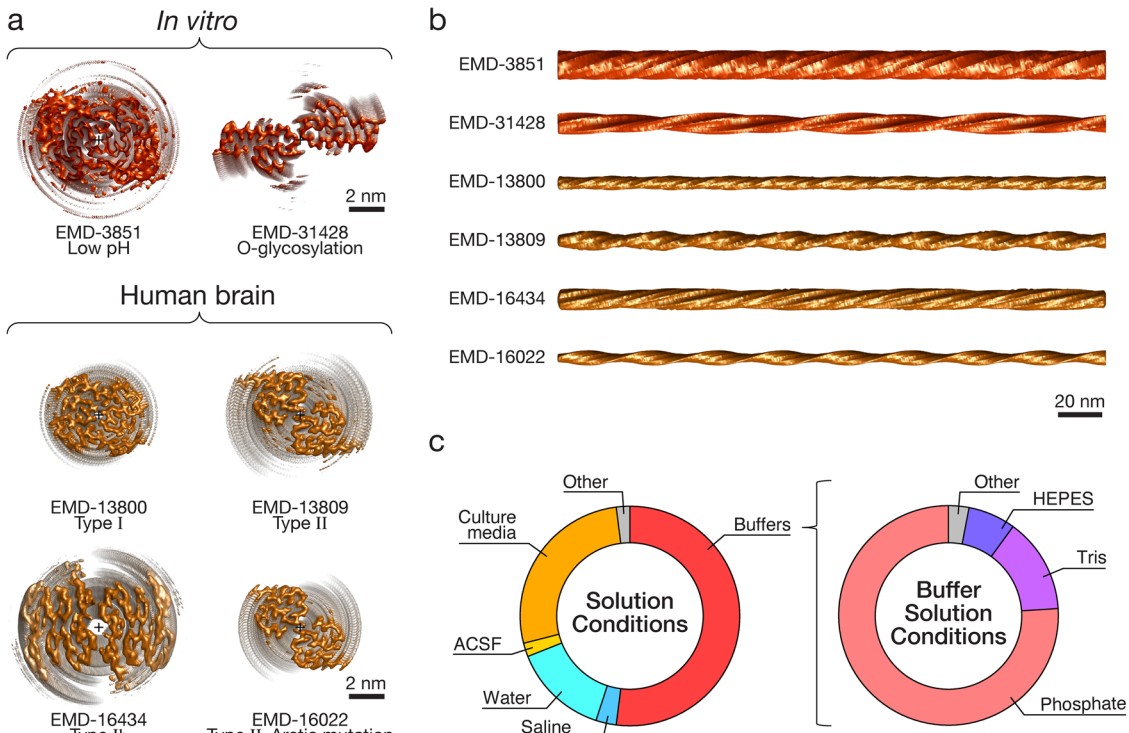

**Fig. 1 | Aβ₄₂ amyloid core structures of human brain derived and in vitro assembled fibrils demonstrate diverse structures and suggest assembly condition-dependent structural polymorphism. a** Examples of recent cryo-TEM derived maps of Aβ₄₂ amyloid filament core deposited in the EMDB with reported resolution of 4 Å or better. The filament cross-sections are shown for two in vitro assembled fibril polymorphs (EMD-3851[21] and EMD-31428[77] coloured in bronze) and the Type I, Ib and II fibril polymorphs seen in human patient brain samples (EMD-13800, EMD-16434 and EMD-13809/EMD-16022[6,24], respectively, coloured in gold). All cross-sections are shown with identical scale, with scale bars indicating 2 nm. **b** The surface envelope of same Aβ₄₂ fibril polymorphs shown in (**a**). The surface envelope structures were generated using axis-aligned and extended cryo-TEM derived maps for comparing the long-range and helical features of the fibril polymorphs[36]. The filaments are shown with identical scale, with scale bar indicating 20 nm. **c** Buffers used in highly cited publications between 2005 and 2020. The most highly cited publications involving Aβ₄₂ from each of these years were tabulated (Supplementary Table S1) and the buffer conditions the monomers was incubated in which substantial polymerisation could occur were recorded. Pie charts to the left display the breakdown of various different types of solution conditions used and to the right the specific buffer salts in buffered conditions used.

with two laterally associated protofilaments. In separate studies with ex vivo seeded samples, an Aβ₄₀ and an Aβ₄₂ structure displayed a left-handed twist while another Aβ₄₂ structure displayed a right-handed twist, all three were composed of two laterally associated protofilaments [22,26]. These studies indicate the potential for different structural variation arising in different local environments, such as different brain regions. However, the range of structural polymorphs Aβ₄₂ amyloid fibrils can adopt has not been mapped.

In order to quantify the structural variation of fibril populations formed from Aβ₄₂ and to map the extent of possible fibril structures that can be formed, *individual* fibrils in a sample population must be resolved at sufficiently high detail to distinguish the structural properties of each individual fibril. Atomic force microscopy (AFM) imaging has previously been used to demonstrate that the distribution of morphometric features can be obtained by the analysis of each individual fibril in a sample (e.g. [27–33]). Recently, we have demonstrated that 3D models of individual fibril structures reconstructed from AFM image data can be used to map the polymorph distribution and to quantify the structural variations within a population[29,34,35]. Here, we obtained detailed images of fibrils formed from recombinant Aβ₄₂ grown in solution conditions with three most prevalently used buffer salts sodium phosphate, HEPES and Tris at neutral pH in vitro and mapped the polymorphic assembly landscape of Aβ₄₂ by applying our individual fibril structural analysis approach. We show that across all the conditions tested, certain classes of fibril polymorphs are conserved. However, the overall diversity of fibril structures *is* sensitive to small changes in the assembly conditions in both the magnitude of the population heterogeneity and the types of filament species formed. Excitingly, we identified rare fibril species in these heterogeneous populations that closely resemble

Aβ₄₂ amyloid purified from human brains by matching individual filament structures to cryo-EM derived structural data[36]. These results suggest that differences in local in vivo environments may lead to distinct sets of fibril structures in heterogenous fibril populations. Therefore, understanding the structures of Aβ₄₂ fibrils formed under different conditions could highlight how different Aβ fibril structures may occur and persist in clinical populations or in different cellular and tissue environments, and these conclusions thrust the biological roles of structural heterogeneity of amyloid populations into focus.

## Results

### Characterisation of purified recombinant monomeric Aβ₄₂

To ensure assembly of high-quality amyloid samples for structural studies of individual fibrils, high-quality monomeric Aβ₄₂ samples were first produced. Production of recombinant monomeric Aβ₄₂ was optimised from previous methods[37,38] to include multi-round size exclusion chromatography (SEC), to ensure high reproducibility of assembly. The final SEC step was performed in multiple rounds until the resulting chromatogram contained a single main peak (Supplementary Fig. S1a) that was collected and used immediately for assembly. SDS-PAGE of the eluted monomer (Supplementary Fig. S1b) further confirmed the purity of the monomer samples. The formation of fibrils from the monomer solution was monitored using Thioflavin T fluorescence (Supplementary Fig. S1c) and CD (Supplementary Fig. S1d), demonstrating the expected concentration dependent sigmoidal kinetics profile and formation of β-rich structures for amyloid assembly. To verify that the samples made using the recombinant Aβ₄₂ protein show the expected toxicity properties, primary hippocampal

neurons from mice were exposed to 10 μM monomer equivalent concentration of oligomeric Aβ$_{42}$ formed during assembly under previously reported conditions[39] and cell viability was assessed using a Live/Dead fluorescence assay (Supplementary Fig. S1e). As expected, cell death was significantly more prevalent in cultures treated with the Aβ$_{42}$ sample ($p < 0.001$), confirming its expected neurotoxicity potential[39]. To evaluate the functional and pathological relevance of amyloid fibrils formed from these Aβ$_{42}$ protein samples, an assay was carried out to monitor the synaptic function of primary mice hippocampal neurons after 24 h incubation of fibrillar Aβ$_{42}$ formed from the protein sample. The results from this experiment (Supplementary Fig. S2) confirm that amyloid fibril samples formed in vitro using the Aβ$_{42}$ samples indeed had a detrimental effect on the synaptic function of the primary neuronal cells. Taken together, these data show that we have generated monomeric protein samples that demonstrate all the biophysical, biochemical and toxic properties expected of assembly competent Aβ$_{42}$ and the fibrils they form.

## Selection of assembly conditions for polymorphic mapping of Aβ$_{42}$

To investigate how solution conditions affect the extent of the polymorphism for Aβ$_{42}$ amyloid assembly, we first identified commonly used solution conditions through an analysis of published experimental Aβ$_{42}$ studies in the literature. In Alzheimer's patients, Aβ amyloid fibrils are found in extracellular plaques in the brain. In the early stages of Alzheimer's, the hippocampal region of the brain is especially vulnerable although as the disease progresses other areas of the brain can be impacted resulting in different symptoms. Different brain regions are likely to have different localised conditions resulting in different Aβ aggregation propensity, behaviour and potentially disease phenotype. This behaviour is also observed in other amyloid fibrils such as tau in which ex vivo structures from patients with different tauopathies have varying structures as determined by cryo-TEM[40].

For in vitro studies, the conditions used to utilise or to resolve Aβ$_{42}$ fibril structures vary considerably between reports, and an array of different Aβ$_{42}$ fibril preparation methods are also employed in the field. In particular, different solution conditions are used to generate Aβ$_{42}$ amyloid samples for molecular, structural, biophysical and cellular studies. We selected the top 20 most cited publications from each of the years (according to Google Scholar and Web of Science) running up to May 2020. Selected publications (presented in a tabulated form in the Supplementary Table S1) were included on the basis that they contained at least one Aβ$_{42}$ experiment in which some form of amyloid assembly had occurred. A quantitative analysis of the buffer conditions used in this literature dataset is shown in Fig. 1c. As seen in Fig. 1c and in Supplementary Table S1, phosphate buffers were the most used (primarily sodium phosphate or phosphate buffered saline which is a mixture of sodium and potassium phosphates weighted heavily towards sodium) although Tris and HEPES buffers were also frequently used. Based on this analysis, to test the effect of solution conditions on the extent of Aβ$_{42}$ fibril polymorphism and the heterogeneity of Aβ$_{42}$ fibril populations, the three most prevalently used buffer salts: sodium phosphate, HEPES and Tris, were selected. Subsequently, Aβ$_{42}$ fibril formation in the presence of these buffer salts was carried out at 37 °C and pH 7.4. In order to compare variation in the extent of the structural polymorphism present in the amyloid fibril population that arises through altering the buffer salts used for amyloid assembly, fibril samples were prepared in an identical manner, except that the final monomer purification steps were performed in the appropriate buffer at pH 8.0 to avoid aggregation on the column[37]. Subsequently, an appropriate solution was added to the monomer to ensure that 20 mM of the buffer salt and a pH of 7.4 were achieved in the final assembly reaction solutions and sodium phosphate buffer at pH 8 were also included in the set of conditions used.

## Assembly of Aβ$_{42}$ result in amyloid fibril samples with high degree of structural heterogeneity

Previous work to determine the structure of amyloid fibrils formed from Aβ has demonstrated their morphology to be highly variable, even for amyloid fibrils assembled under identical conditions in the same sample or when extracted from Alzheimer's patients[6,40–42]. Therefore, to understand how Aβ$_{42}$ amyloid structures link with disease aetiology, the assembly landscape relating to the fibril polymorphs that can form must first be mapped and understood.

In order to map the polymorphic assembly landscape of Aβ$_{42}$, we next employed peak-force tapping mode AFM imaging (ScanAsyst imaging mode, Bruker), which allows for topographical imaging with a high level of control over the imaging force applied to avoid imaging-force dependent compression and deformation of the filament structures that would result in lower apparent height measurements. Low magnification AFM images shows that the Aβ$_{42}$ fibrils were dispersed randomly in terms of their spatial and orientational spread, with some noticeable lateral association between filaments as well as some bundling of fibrils (Fig. 2a). Qualitatively, the overall suprastructural appearance of the fibrils show subtle differences in HEPES and Tris buffers compared to the samples in phosphate buffers. In phosphate buffer, particularly at pH 8.0, fibrils appeared as more tightly packed bundles of fibrils than in HEPES pH 7.4. This may reflect filament-filament interactions under the buffer conditions employed but could also be due to the conditions employed during deposition of the filament on the mica support. Upon magnification, the variety in fibril structures formed under different experimental conditions is striking (Fig. 2b). In addition, it was not uncommon to observe fibrils which progressed from one type of polymorph to another across the length of a single fibril often with a noticeable change in average height and twist pattern. This intra-fibril variation could be due to attachment or detachment of additional protofilament or fibril fragments, and strongly supports the view that variation in protofilament assembly and organisation[43] is an important molecular mechanism of amyloid assembly and polymorphism[44]. In summary, considerable structural variations and heterogeneity are observed in all samples across all conditions tested. These data demonstrate that Aβ$_{42}$ fibril assembly have a high propensity for polymorphism and produce highly heterogeneous fibril populations.

## Quantification of structural variation in Aβ$_{42}$ fibril populations reveal high degree of structural polymorphism that is sensitive to assembly conditions

To quantitatively assess and compare the structural variation in the fibril populations, we next analysed the three-dimensional structures of individual filaments reconstructed from the topographical AFM images using the contact point reconstruction (CPR-AFM) method described previously[34,45]. Recent developments in individual filament 3D-reconstructions of amyloid fibrils formed from a tau sequence[36,46], and from a β$_2$-microglobulin variant[47], by AFM carried out in parallel with cryo-TEM, has validated that the reconstructed 3D surface envelopes of the individual filaments using AFM height images indeed match with the cryo-TEM derived maps of identical filament species. Here, Individual fibrils (those with segments which are clearly distinguishable from a bundle of fibrils) with no overlap with other fibrils, and with at least 3 repeating cross-overs and 150 nm in uninterrupted length, were selected. One hundred such fibrils from each of the four assembly conditions were traced and their individual 3D surface envelopes were reconstructed for subsequent analysis. Digitally straightened fibril images, reconstructed 3D surface envelope structural models and cross-sectional contact point density maps, as well as the morphometric parameters for each of the 400 fibrils resolved by AFM can be found in the Supplementary Figs. S3, S4, S5, and Supplementary Table 2, respectively. This analysis revealed varying helical twist and widths of the individual filaments (Supplementary Fig. S4), as well as their diverse cross-sectional size and shape (Supplementary Fig. S5). This is consistent with varying protofilament folds and number of protofilaments[43] within and across the individual fibrils in the fibril populations. The average fibril height distributions of the fibril populations formed in each of the assembly conditions (Fig. 3a) suggest differential heterogeneity of the populations despite the similarity in the assembly conditions used. The distinctive spread of fibril structures formed in each of the different conditions used is unequivocally

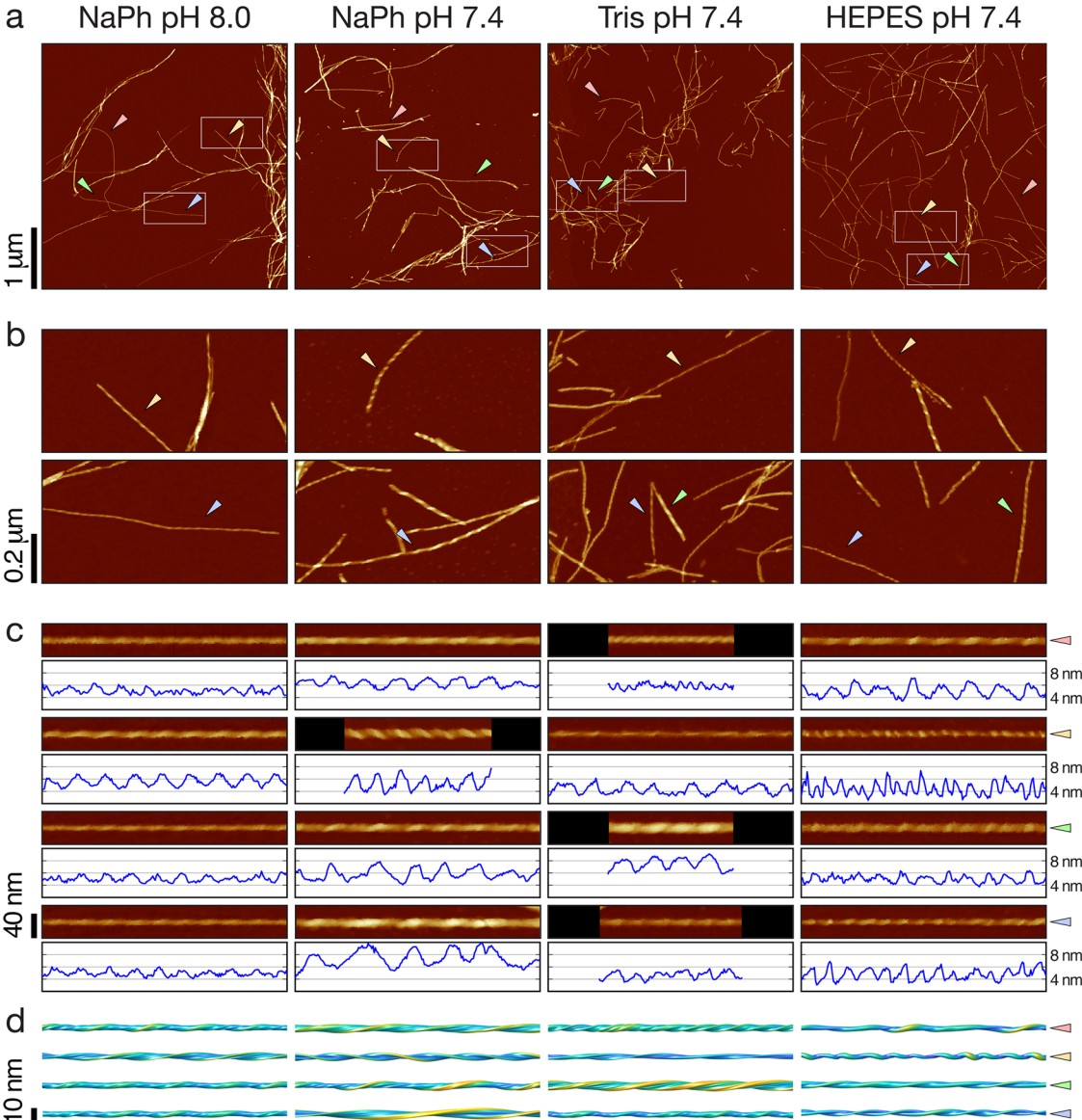

**Fig. 2 | Topographical AFM height images of Aβ₄₂ fibrils reveal heterogeneous and polymorphic fibril populations formed under different experimental solution conditions.** AFM height images of $A\beta_{42}$ fibrils formed in 20 mM sodium phosphate (NaPh) pH 8.0, NaPh pH 7.4, Tris pH 7.4 and HEPES pH 7.4, respectively. **a** AFM height images of 4 × 4 μm areas. Qualitatively, all four types of samples appear to generate a variety of different structural polymorphs in heterogeneously mixed populations within each sample. **b** 4× magnified height images representing 1 × 0.5 μm areas shown in the white rectangles in (**a**). The different structural polymorphs are clearly visible at this magnification, including different twist patterns and height profiles. **c** Examples of individual fibrils traced from the images in (**a**). The individual fibrils are indicated by coloured triangles in (**a**, **b**) and up to a 400 nm digitally straightened segments are shown together with their central-line height profiles. **d** 200 nm segments of 3D surface envelopes reconstructed from the same fibril images shown in (**c**), demonstrating the diversity of structures observed in the samples. In all AFM images, the colour scale represents a height range from −5 to 10 nm.

demonstrated by the 2-dimensional (2D) contour maps of average fibril heights against the directional periodic frequency ($dpf$ = −1/helical pitch for left-hand twisted filaments or +1/helical pitch for right-hand twisted filaments[29]), which allows visualisation of the distinct polymorph distributions in each of the conditions (Fig. 3b). Height and $dpf$ contour map analysis, in particular, demonstrated how the fibril populations varied between the different samples. Fibrils formed in phosphate buffer, particularly at pH 8, tended to form wider filaments with a lower frequency of twist, likely promoted by a higher frequency of inter-protofilament association (Fig. 3b). In contrast, Tris buffered reactions at pH 7.4 produced a greater range of twist patterns than the other conditions tested, which may reflect modulation of inter-protofilament interactions. Lesser variation in average height was also observed in Tris buffered reactions compared with phosphate buffered assembly. Assembly in HEPES buffer, on the other

hand, produced the overall least variation in fibril width and twist compared to the other conditions tested. Both left and right handedness of twist is observed in the data, both of which have been observed in the four different Aβ samples. However, right-hand twisted fibrils remain rarely observed across all four assembly conditions, accounting for only 10.3% (41) of the 400 fibrils analysed (Supplementary Fig. S6). To investigate how these polymorph distributions form and evolve during the assembly of the fibril populations, we monitored the polymorph distribution during de novo amyloid assembly reactions in the HEPES buffer condition (Supplementary Fig. S7, and Supplementary Table S3). At the 1-h timepoint where the majority of monomeric assembly precursors have already depleted (red ThT kinetics traces in Supplementary Fig. S1c), thin filaments that display a large variation in their helical twist have assembled. After 6 h, the wider fibrils that were seen in the polymorph distributions (Fig. 3) started to appear alongside

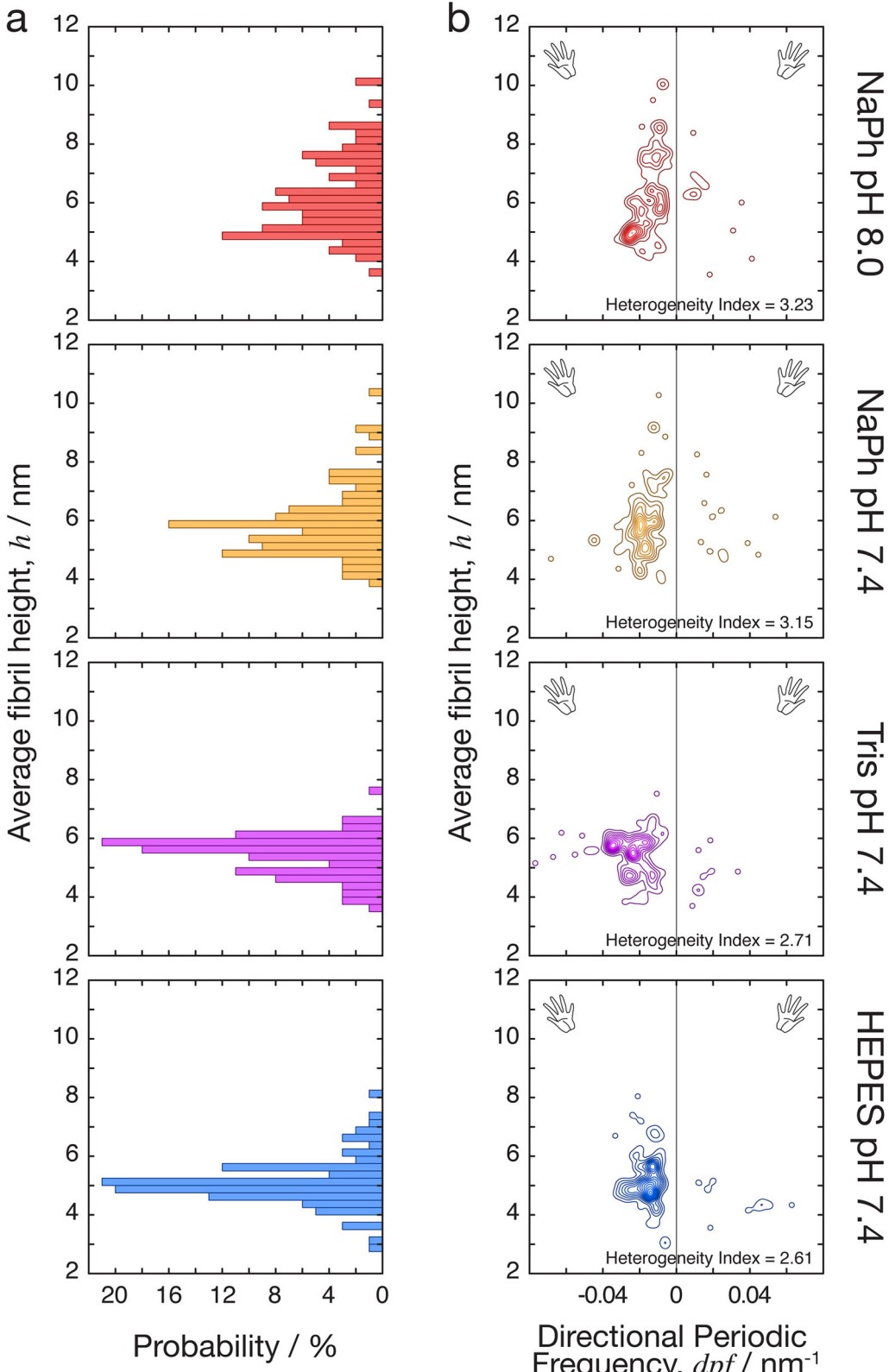

**Fig. 3 | The distribution of structural polymorphs of Aβ$_{42}$ fibrils is sensitive to the assembly solution condition.** Morphometric analysis of Aβ$_{42}$ fibrils formed in 20 mM sodium phosphate (NaPh) pH 8.0, sodium phosphate pH 7.4, Tris pH 7.4 and HEPES pH 7.4, respectively. **a** The distribution of average fibril height for the heterogeneous fibril populations formed in each of the assembly conditions tested. **b** Contour maps displaying average height of individual fibrils (describing their cross-sectional width) plotted versus their individual *dpf* values (describing their

helical properties of handedness and pitch). The structural variation of Aβ$_{42}$ fibrils formed in each of the four assembly conditions are visualised and compared through these contour maps by spreading the characteristics of their structures onto two dimensions. The heterogeneity index, HI (larger values indicate higher heterogeneity as defined in the 'Methods' section), for each of the population distributions is also shown.

the species seen at the 1-h timepoint. The polymorph distribution settled into a distribution comparable to that seen in Fig. 3 at the 168-h timepoint. Interestingly, at one the 1-h timepoint, the thin fibril populations showed a substantial subset with right-hand twisted fibrils (Supplementary Fig. S7). These thin filaments were not observed at later time points, suggesting that assembly of $A\beta_{42}$ amyloid fibrils proceed through an intermediate polymorph distribution, with individual filaments required to either disassemble and reassemble into wider filaments, undergo a protofilament assembly process, or both[48,49]. These time-dependent data also demonstrate that the assembly of the diverse fibril populations captured by the polymorph distribution maps seen in Fig. 3 occurs on a slower timescale than the depletion of free monomers in solution indicated by the onset of the plateau phase in ThT kinetics traces (Supplementary Fig. S1c). Overall, these polymorph distribution maps suggest that both the magnitude of heterogeneity and the types of filament species formed are sensitive to assembly conditions. This is especially remarkable considering the apparent similarity of the assembly conditions used.

To visualise, organise and identify natural partitions within the individual-fibril level structural data that may represent distinct classes of fibril structures, we carried out an agglomerative hierarchical clustering analysis[29]. For this structure based clustering analysis, we developed a distance function $d_\xi$ (see 'Methods') that takes into account the detailed cross-sectional shape[36] as well as the helical symmetry properties described by $dpf$ parameter[29]. This pairwise $d_\xi$ distance score, where low numerical value indicate high degree of structural similarity and high numerical value indicate dissimilarity for a pair of individual fibrils, were calculated for all 79,800 pairings possible for 400 individual fibril structures analysed. This approach allowed us to quantify and objectively compare the heterogeneity of each of the four fibril populations, and to detect if the fibril structures can be divided into classes that correlate with the assembly conditions.

For quantitative comparison of the magnitude of heterogeneity of the four fibril populations, we defined a heterogeneity index: $HI = RMS_{d_\xi}$, based on the quadratic mean of $d_\xi$ distance for all fibril pairs within each sample population. The HI value of the observed fibril population at each assembly condition was subsequently calculated and ranked objectively, revealing the order of high to low heterogeneity: sodium phosphate buffer at pH 8 (HI = 3.23 ± 0.15 SE), sodium phosphate buffer at pH 7.4 (HI = 3.15 ± 0.19 SE), Tris buffer at pH 7.4 (HI = 2.71 ± 0.28 SE) and HEPES buffer at pH 7.4 (HI = 2.61 ± 0.19 SE). These results corroborated the qualitative assessment of fibril population heterogeneity seen in the height vs. $dpf$ contour map analysis (Fig. 3b).

In terms of the species present in each of the four fibril populations, the agglomerative hierarchical clustering analysis of the entire dataset of 400 fibrils using the pairwise $d_\xi$ distance measure (Fig. 4a) demonstrated that fibrils of similar structures that may constitute a distinct fibril polymorph class can often be found in more than one condition examined. This can be seen in the colour bar to the right of the dendrogram in Fig. 4a, showing that each cluster in the data with distance $d_\xi < 1$ is almost always composed of individual fibrils observed from two or more assembly conditions. This result, again, corroborates with the qualitative assessment of the fibril populations seen in the height vs. $dpf$ contour map analysis (Fig. 3b) where there is a considerable overlap of fibril structures observed in a region of the maps with left-handed twist with 10–40 repeating units per μm (0.01–0.04 repeating units per nm) and an average fibril height between 4–6 nm across the different conditions tested. Whereas the coarse contour map analysis is not able to separate fibril structures with further detail than their width and helical morphometrics, the hierarchical clustering analysis of 3D-reconstructed filament envelopes is able to analyse and cluster filaments based on their cross-sectional shapes in addition to their helical pitch and handedness (Supplementary Fig. S4 and S5). Thus, this analysis suggests that some fibrils of similar structures that may constitute a common fibril polymorph were able to form in some or all of the buffer salt or pH conditions examined. Overall, these results allow us to confirm, in a quantifiable manner, that $A\beta_{42}$ fibrils are able to form a cloud of different structural polymorphs under identical conditions in the same sample, and that each

and every fibril displays structural individuality, as previously seen in the assembly of short amyloid forming peptides[29]. These results also confirm the observations made in earlier works that $A\beta_{42}$ amyloid fibrils are highly polymorphic, and the amyloid sample populations are highly heterogeneous in vitro and ex vivo from patients[6,16,22].

## The in vitro population cloud of polymorphic $A\beta_{42}$ structures encompasses fibril structures found from ex vivo human patient brains

Finally we investigated if the heterogeneous $A\beta_{42}$ fibril populations we observed encompasses amyloid fibril structures that can be found in humans. There are now several different experimentally determined structural models describing the filament core structures of $A\beta_{42}$ fibrils. In particular, recent reports detail cryo-TEM based structure determination that revealed the core structures of several $A\beta_{40}$ and $A\beta_{42}$ fibril polymorphs from brain samples of human patients suffering from AD and related dementias in atomic detail[6,22–26]. In order to validate whether the maps of the polymorphic $A\beta_{42}$ assembly landscape we observed encompasses known $A\beta_{42}$ polymorphs characterised to high detail by cryo-TEM, we calculated the distance score, $d_\xi$ (see 'Methods'), between every one of the 400 fibril structures we 3D-reconstructed and analysed by AFM, and all of the recent $A\beta_{42}$ cryo-TEM maps (Table 1) deposited in the Electron Microscopy Data Bank (EMDB). We also included recent cryo-TEM maps of $A\beta_{40}$ fibrils found in the EMDB as controls since current evidence strongly shows that fibrils formed of $A\beta_{42}$ are different to those of $A\beta_{40}$ (Table 1). Comparison and structural distance calculations between the Coulomb potential maps from cryo-EM experiments in the EMDB and the individual filament envelopes from our AFM data was accomplished by modelling the rigid-body interaction of an AFM tip of average sharpness used in our experiments with each of cryo-TEM map to generate simulated AFM images of the fibril polymorphs seen by cryo-TEM (Fig. 5a). The shapes of the external envelope of the fibrils were determined by calculating the contact points between a simulated AFM tip of average sharpness and the iso-surface of the cryo-TEM derived structural maps with author-recommended contour levels, which are then denoised, axis-aligned, and extended using the reported twist and rise information in each of the EMDB entries[36]. These simulated images demonstrate the considerable variety of the $A\beta$ structures and the considerable morphometric differences between each. The twist and rise information was also used to calculate the $dpf$ values of each cryo-TEM derived data entry. In Fig. 4b, a scatter plot of these data is overlaid onto the assembly contour map from all the individual fibrils formed in the Tris buffered condition used here. Interestingly, many cryo-TEM derived entries, including the $A\beta_{40}$ entries, do not fall within the contour boundaries, but Type I and Type II $A\beta_{42}$ polymorphs observed from human brain samples were located well within the mapped polymorph distribution contour boundaries.

To individually match the 400 AFM-derived 3D-reconstrructed fibril structures to the cryo-TEM derived data (Table 1), the $d_\xi$ distance values were calculated and ranked for all 8400 pairwise comparisons between the 21 cryo-TEM derived structural maps and 400 individual filament envelopes reconstructed from our AFM data (Fig. 4c). Strikingly, of the top 3 out of the 8400 ranked pairs showing the highest pairwise similarities with lowest $d_\xi$ distance scores below 0.5, two are individual fibrils that match to the Type II polymorph structure and one is an individual fibril that matches to the Type I polymorph structure of $A\beta_{42}$ seen in the brain samples from human patients[6,23,24] (arrowheads in Fig. 4c). The clusters of fibrils (defined in the 'Methods' section) containing the top matched individuals (Fig. 4a blue and purple clusters matching to Type I and Type II human brain $A\beta_{42}$ polymorphs, respectively) are all ranked in the top 20 for each matched cryo-TEM entry. The individual fibril and its parent cluster that matches to the Type I polymorph (Fig. 4a blue cluster) account for 2.25% (9 out of 400) and the two individual fibrils and their parent cluster that matches to the Type II polymorph (Fig. 4a purple cluster) account for 1.25% (5 out of 400) of the individual fibril observations. The same analysis was carried

**Fig. 4 | Comparison of individual Aβ$_{42}$ fibril structures observed by AFM to cryo-TEM derived structural maps suggest rare fibril structures in the heterogeneous populations match fibril polymorphs seen in human patient brain samples. a** Agglomerative hierarchical clustering analysis of all 400 individual fibrils analysed from the four assembly conditions. This analysis organises the fibrils based on their individual surface envelopes obtained from 3D reconstructions using individual filament level AFM data, with the order of the filament from bottom to the top of the dendrogram matching the order of individual fibrils in Supplementary Figs. S3, S4 and S5. Colour bar to the right of the dendrogram indicate the origin of each of the filaments in terms of their assembly condition, indicating that the fibril polymorphs were generally observed across the experimental assembly conditions tested. **b** Contour maps displaying average AFM tip-accessible cross-sectional area of individual fibrils (*csa*, describing their cross-sectional area) plotted versus their individual *dpf* values (describing their helical properties of handedness and pitch) for the polymorph distribution of fibrils assembled in Tris pH 7.4. The cross-sectional area (*csa*) and the directional periodic frequency (*dpf*) of cryo-TEM derived Aβ$_{40}$ and Aβ$_{42}$ structural data entries in the EMDB are also mapped onto the contour maps for comparison. Type I and Type II Aβ$_{42}$ fibril polymorphs from human brains[6] are highlighted by light blue and pink circles, respectively. The bottom contour map is a zoom-out of the region represented by top map highlighted by black square. **c** The similarity between each of the 400 individual fibrils and the cryo-TEM derived structural maps quantified by the pairwise $d_\xi$ distance scores. All of the pairwise scores are visualised by the colour scale shown to the bottom left, with low $d_\xi$ distance scores indicating high similarity. The rows of this image represent each of the 400 fibrils in the same order as shown in the dendrogram in (**a**) and the columns represent an EMDB-entry with their identity labelled at the bottom. The identity of known Aβ$_{42}$ fibril polymorphs purified from human brains are also indicated. Arrow heads show the top three ranked matches, and the light blue or pink colour of the arrow heads and clusters refer to matches with Type I and Type II Aβ$_{42}$ fibril polymorphs[6], respectively.

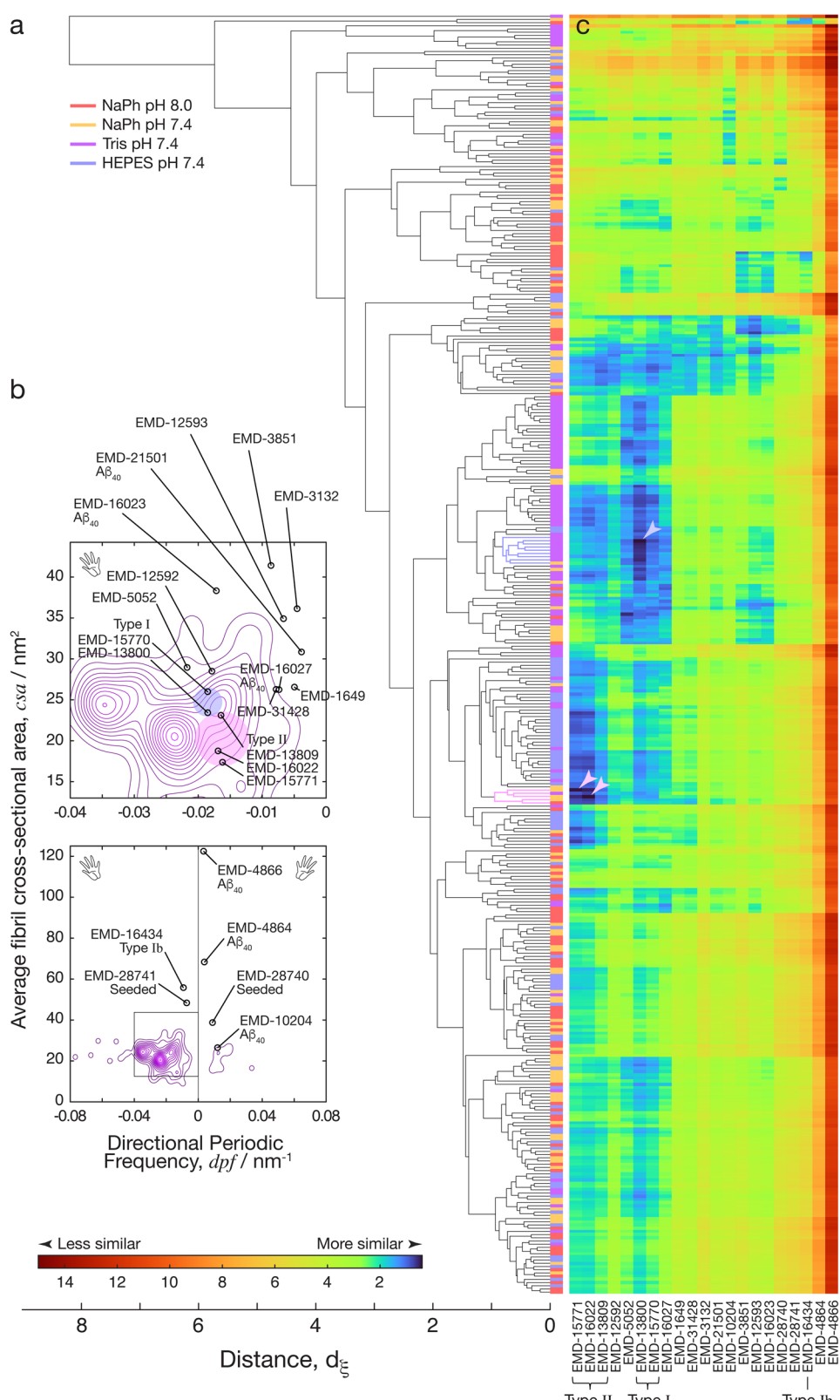

out using 4000 simulated AFM fibril data obtained with randomly generated cross-sectional and helical properties (Methods) confirmed that the probability of obtaining a positive match by chance is very small compared to the observed frequency for the matched clusters of rare species. Thus, these clusters represent rare sub-populations within the heterogeneous Aβ$_{42}$ amyloid fibril population clouds formed in vitro

under the assembly conditions used that closely resemble Aβ$_{42}$ fibril structures found in human patient brain samples.

To further validate the structural similarity between the rare fibril species found in the heterogenous in vitro assembled fibril populations and the Aβ$_{42}$ fibril structures from human patient brain samples, the top matched individual fibril seen with AFM were further validated by comparing

**Table 1 | Cryo-TEM structural data entries of polymorphic Aβ amyloid fibrils in the EMDB**

| EMDB ID | 40/42 | Fibril type | Resolution/Å | Year Released | Publication |
|---|---|---|---|---|---|
| EMD-1649 | Aβ42 | in vitro | 15 | 2009 | Schmidt et al.[19] |
| EMD-5052 | Aβ42 | in vitro | 10 | 2010 | Zhang et al.[20] |
| EMD-3132 | Aβ42 | in vitro | 7 | 2015 | Schmidt et al.[18] |
| EMD-3851 | Aβ42 | in vitro | 4 | 2017 | Gremer et al.[21] |
| EMD-12592 | Aβ42 | in vitro, AβUpp1-42Δ19-24 | 5.7 | 2021 | Pagnon de la Vega et al.[76] |
| EMD-16434 | Aβ42 | Human brain | 3.5 | 2022 | Yang et al.[6] |
| EMD-12593 | Aβ42 | in vitro, AβUpp1-42Δ19-24 | 5.1 | 2022 | Pagnon de la Vega et al.[76] |
| EMD-31428 | Aβ42 | in vitro, dsAβ42 Tyr10 O-glycosylation | 3.1 | 2022 | Liu et al.[77] |
| EMD-28740 | Aβ42 | Human brain, seeded | 2.83 | 2022 | Lee et al.[22] |
| EMD-28741 | Aβ42 | Human brain, seeded | 2.76 | 2022 | Lee et al.[22] |
| EMD-15770 | Aβ42 | Human brain | 2.9 | 2022 | Stern et al.[23] |
| EMD-15771 | Aβ42 | Human brain | 3.7 | 2022 | Stern et al.[23] |
| EMD-13800 | Aβ42 | Human brain | 2.5 | 2022 | Yang et al.[6] |
| EMD-13809 | Aβ42 | Human brain | 2.8 | 2022 | Yang et al.[6] |
| EMD-16022 | Aβ42 | Human brain, E22G | 2.8 | 2023 | Yang et al.[24] |
| EMD-10204 | Aβ40 | Human brain, meninges | 4.4 | 2019 | Kollmer et al.[25] |
| EMD-4864 | Aβ40 | Human brain, meninges | 5.56 | 2019 | Kollmer et al.[25] |
| EMD-4866 | Aβ40 | Human brain, meninges | 7.01 | 2019 | Kollmer et al.[25] |
| EMD-21501 | Aβ40 | Human brain, seeded | 2.77 | 2021 | Ghosh et al.[26] |
| EMD-16023 | Aβ40 | Human brain, E22G | 1.99 | 2023 | Yang et al.[24] |
| EMD-16027 | Aβ40 | AppNL—G—F mouse brains, E22G | 3.5 | 2023 | Yang et al.[24] |

Structural maps of Aβ42 amyloid fibrils from 2009 and recent Aβ40 amyloid fibrils from 2019 with resolution around 7 Å or better in the EMDB included in this study are listed in the order of release year.

their fibril images with all of the simulated images made using cryo-TEM derived structural maps (Fig. 5a). The cross-sectional contact-point density maps of the best matched individuals were also overlaid with the cross-sections of cryo-TEM derived structural maps of Type I and Type II Aβ42 amyloid from human brain samples and their fitted molecular models, respectively (Fig. 5b–e), demonstrating the remarkable similarities. Interestingly, no other of the 400 individually analysed fibril structures by AFM matched well with any other cryo-TEM derived structural maps included in the analysis. Importantly, no Aβ40 structural matches were identified. Therefore, despite that only the surface envelopes of the individually analysed fibrils were reconstructed and matched, these results taken together demonstrate strong evidence suggesting that the synthetic fibrils formed in vitro from recombinant Aβ42 can represent, in rare parts, disease relevant fibril structures formed in vivo. These conclusions demonstrate that a common set of polymorphs, including disease related polymorph structures, do populate and overlap across the different condition in vivo and in vitro despite the sensitivity of the assembly landscape.

## Discussion

When compared to conventional protein folding, in which structural polymorphism is rare, amyloid assemblies demonstrate a remarkable ability to have a single sequence of amino acids fold and assemble in numerous ways to make up the cross-β core regions of fibrils. One clear example is the range of ex vivo fibril structures reported from various patients with different diseases in which tau fibril formation is reported[40]. Structural variation could be an important factor in amyloid disease as distinctive structures could arise from different local conditions, requiring different treatment options dependent on the local environment and the structure formed in that disease[50]. Equally, different structural polymorphs could each have a distinctive set of biophysical and biochemical properties, display dissimilar surfaces and therefore contribute to the local biological environment in different ways, therefore having diverse impacts in different diseases. Finally, structural variation itself may result in a dynamic and continuously evolving cloud of fibril species which could result in a variety of behaviours and can

change upon changes to the local environment, making different disease states difficult to predict and to target. It is already known that Aβ can form polymorphic structures which has been demonstrated in both in vitro and ex vivo samples (Table 1). However, the polymorphic extent and the heterogeneity of Aβ fibril assembly has not been quantified to establish the ruggedness of the landscape of polymorphic Aβ assembly. Furthermore, the effect of changing the assembly conditions on Aβ polymorphism has not previously been quantified despite a plethora of different conditions were used in the literature to date (Fig. 1c and Supplementary Table 1). Finally, it remains hitherto unclear as to whether a subset of polymorphs is conserved across conditions in vitro and in vivo or whether changing the assembly conditions resulted in an entirely different set of structures.

Due to its importance in AD, Aβ42 is a highly researched amyloidogenic peptide sequence. In terms of the amyloid fibril structures, numerous different structural polymorphs have been determined from various cryo-TEM and ssNMR studies. Rather than adding evidence to confirm and to improve previous structural models of Aβ42 amyloid, each new study has often revealed entirely new structures, despite investigating the same primary amino acid sequence. This has cemented the idea that Aβ42 amyloid assembly can be highly polymorphic. Differences in the assembly conditions used in vitro have often been cited as contributing factors towards the variations in experimental results. Unsurprisingly, different experiments can often require different experimental conditions. For example, the strength and integrity of the signal obtained using circular dichroism is highly dependent on the solution used[51]. Differences in the type of experiment being performed, along with research group specific preferences have resulted in a vast array of experimental conditions in which Aβ42 polymerisation has been studied (Supplementary Table S1). Therefore, in this work, we sought to understand how different experimental conditions contribute to the structural differences of the fibrils generated as well as the heterogeneity of the fibril populations formed. To do so, we have collated commonly used experimental conditions from highly cited publications from the last 15 years involving Aβ42 assembly and developed a unique method to quantify, by structural analysis at individual fibril level, the extent

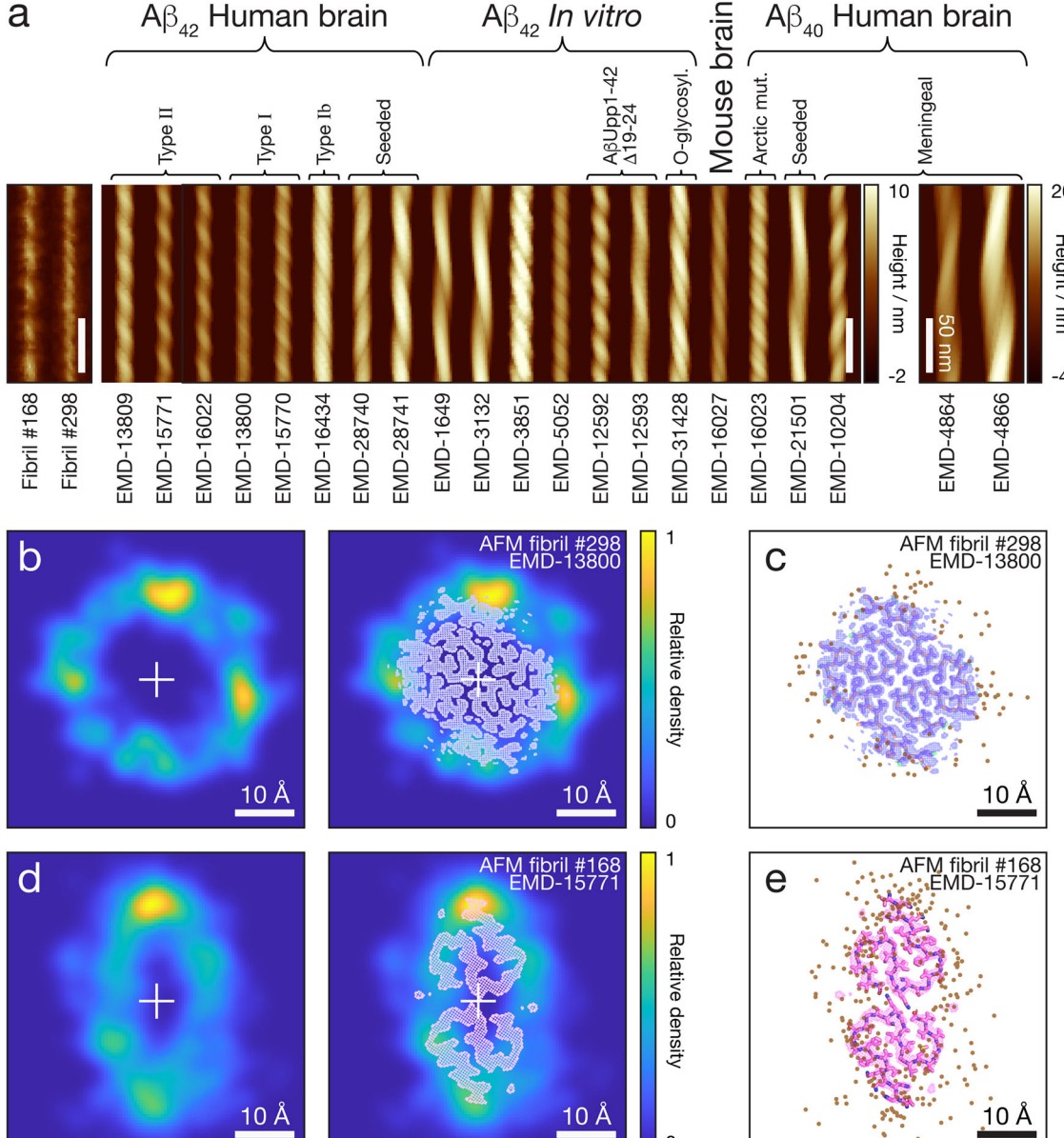

**Fig. 5 | Structural comparison of Type I and Type II Aβ₄₂ fibril polymorphs from human patient brain samples with their respective best matched individual fibrils seen in the heterogeneous fibril populations. a** Comparison of the straightened AFM height image of the two Aβ₄₂ fibrils best matched to cryo-EM derived structural maps with simulated topographical AFM images of Aβ amyloid fibril polymorph data entries found in the EMDB. The Aβ fibril polymorph images were simulated with a tip radius of 5.5 nm matching to the average tip radius estimate of the AFM data using the iso-surfaces of axis-aligned and extended cryo-EM structural maps. A 180 nm segment is shown for each fibril image and the scale is identical for all images except EMD-4864 and EMD-4866 due to their large cross-sections as indicated by their separate colour scale. The scale bars indicate 50 nm in all images. The centre-line sections of the two best matched fibrils compared with that of the simulated best-match AFM images are also show for comparison in Supplementary Fig. S8. **b** AFM cross-sectional contact-point density map of the Aβ₄₂ fibril best matched to Type I polymorph from human patient brains[6]. The AFM-derived density map is shown with and without the cross-section of cryo-TEM derived structural map superimposed for comparison. **c** The fibril cross-section of the Type I polymorph from human patient brains shown as cryo-TEM Coulomb potential map superimposed with the fitted structural model (PDB: 7Q4B) together with the cross-sectional coordinates of the reconstructed AFM tip-filament contact-points (dots) along the top of the fibril contour deconvoluted and 3D-reconstructed from the topographical AFM data. The RMSD between the tip-accessible cross-sectional cryo-TEM derived iso-surface and AFM-derived surface envelope is 1.38 Å. **d, e** The same as (**b, c**) but for the Aβ₄₂ fibril best matched to Type II polymorph from human patient brains (EMD-15771 and PDB: 8BFZ[6]) with a RMSD between the tip-accessible cross-sectional cryo-TEM derived iso-surface and AFM-derived surface envelope of 1.65 Å. All scale bars in (**b–e**) indicate 10 Å in length.

of structural polymorphs that arose when forming fibrils using the different conditions adopted by the field.

We have observed a highly polymorphic landscape of structural possibilities for Aβ₄₂ fibril formation which is not simply constrained to one or few types of structural polymorph per assembly condition. Both right and left-hand twisted fibrils were also observed under all experimental conditions although left-handed fibrils account for around 90% of the individual

filaments we analysed (Supplementary Fig. S6). Whilst phosphate buffers appeared to promote variation in fibril width, the Tris buffered condition promoted variations in the helical twist. These data have confirmed that Aβ₄₂ fibril samples are highly polymorphic, and its rugged polymorphic landscape of assembly is surprisingly sensitive to assembly condition, as the polymorph distribution can be shifted by small changes to the assembly conditions both in terms of the species formed and the overall heterogeneity

of the population distribution. Thus, the approach presented here to resolve the amyloid polymorph distributions will unlock future experimental opportunities on characterising (e.g. the seeding, toxicity and strain-like behaviour), enriching, or targeting specific, potentially rare, amyloid polymorph species.

The supra-structural features of amyloid fibril populations, including both the individual fibril level structural parameters and the population level interconnectivity, are likely to impact upon their biological activity. For example, filaments with a larger surface area to volume ratio could be more efficient for secondary nucleation or other surface catalysed reactions resulting in the formation of toxic oligomeric species[52]. Thin fibrils with a low frequency of twisting may be susceptible to breaking and therefore fragment more frequently, which is known to increase the rate of amyloid uptake into cells[53]. Indeed, different polymorphic structures resulting from the same amyloid forming protein may considerably alter the fibrils' growth, division by fragmentation and chaperone interactions[54–56]. Here, we observe variations in A$\beta_{42}$ fibrils' supra-structural features that are dependent on the experimental conditions. Phosphate buffer, in particular at pH 8.0, promotes a higher degree of inter-protofilament association and ultimately thicker and wider fibrils. Since phosphate is a polyprotic base, it could provide greater local electrostatic shielding between filaments compared to the other two buffer salts at the same molar concentration, resulting in some thicker fibrils, similar to the way in which high ionic strength solutions have been reported to shield electrostatic repulsion between monomers and fibrils in turn catalysing secondary nucleation effects[57]. Fibrils formed in Tris buffer displayed more variable twist patterns but lesser variation in average height between fibrils. This could be due to perturbed intra-filament interactions resulting in no single favourable periodicity in these conditions, which in turn could result in fewer thicker filaments as favourable orientations for inter-filament interactions may be reduced. In this context, secondary nucleation is likely to be impacted by the molecular surface differences due to the supra-structural features of the fibril populations. Here, our data indicate that the supra-structure, including structural variation between filaments and heterogeneity of the fibril population, is sensitively influenced by the experimental conditions. Therefore, it is reasonable to assume that differing conditions, for example in different brain regions, can alter the pathological contribution of A$\beta_{42}$ fibrils by means of changing their supra-structures.

The toxic potential of A$\beta_{42}$ amyloid species is thought to be size dependent. Typically, smaller species or small fibril fragments rather than large fibrils or plaques are considered to more acutely toxic[3,32,39,58]. Furthermore, a general feature of amyloid fibrils is that fragmented fibrils can produce toxic effects[32,59,60] although intermediate oligomeric species of A$\beta_{42}$ may yet possess a more potent cytotoxic potential[61]. These cases highlight the potential for A$\beta$ fibrils to act as a long-term source of small species with cytotoxic potential. A$\beta_{42}$ fibrils can also induce a positive feedback loop of increased toxic potential through secondary nucleation events that results in smaller species capable of toxic effects[62]. However, the relationship between the structure of A$\beta_{42}$ fibril polymorphs and their function with respect to their individual capacity for fragmentation or catalysing secondary nucleation is not well understood. Amyloid fibrils have been suggested to show prion-like properties in which they propagate from cell to cell[63–65]. Examples of this behaviour is analogous to epigenetic gene regulation in yeast by yeast prions[64,66] and the spread of toxic particles in various prion diseases such as Creutzfeldt-Jakob disease[67]. A$\beta_{42}$ has also been shown to be able to spread in a prion-like manner[68]. Prion structures that spread can have a structural dependence, resulting in some cells populated with different 'strains' of fibrils[63–65,69,70]. Structural polymorphism can therefore have an impact on the prion-like behaviour as higher variation in the fibril populations could result in a higher likelihood of fibrils proliferating with structural features which promote fragmentation and/or spreading. Smaller particles, for example, have been observed to link with the infective potential of prion particles[53] indicating that having at least one subset of structural polymorphs which are prone to fragmentation in a heterogeneous polymorph distribution may increase the infective potential of a population of amyloid fibrils. Hence, using experimental conditions that encourage populations of fibrils that have a thin, easily broken morphology could result in significantly different outcomes in experiments measuring the spread of A$\beta_{42}$ between cells.

Amyloid fibrils from the same precursor protein can be implicated in numerous diseases. One example of such is Tau which is implicated in numerous tauopathies, including in AD. Cryo-TEM of ex vivo Tau fibrils from patients with different tauopathies has demonstrated that the core structure of tau fibrils varies in a disease dependent manner[40]. From current structural evidence, it is not yet clear if A$\beta$, which is also associated with several different diseases as well as AD, demonstrates similar behaviour compared to tau in forming disease specific structures, or if A$\beta$ fibrils formed in different brain regions form specific structures. It is also not clear if the lower levels of heterogeneity and structural variations in amyloid populations seen in the structural studies of ex vivo brain samples (Table 1) compared to the high level of structural variations observed here are due to specific difference in the in vivo biological environments, long timescale properties of the assembly mechanism or methodologically linked observation biases. For example, a recent study where A$\beta$ fibril plaques from mouse brains were imaged in situ by cryo-electron tomography revealed that, whilst fibrils extracted from the brains had the same atomic structure solved by cryo-TEM as those previously identified, in situ the observed fibrils were quantifiably thinner indicating the presence of a different polymorph that existed in vivo[71]. In early-stage AD the hippocampal region is considered to be particularly at risk before other regions of the brain become impacted in later stages[72]. Structural features such as the ability to fragment and spread or act as a catalysing surface for secondary nucleation or other surface interactions could contribute to the biological behaviours of amyloid populations propagating from early stages to late stages of disease. Structural variation in A$\beta$ fibrils could increase the likelihood of fibrils that are more prone to spreading resulting in the deposition of A$\beta$ plaques found throughout the brains of patients with late-stage AD.

In summary, we have demonstrated that A$\beta_{42}$ amyloid fibril populations are highly polymorphic and sensitive to assembly conditions. We have also demonstrated that by applying AFM imaging and individual filament 3D structural analysis by CPR-AFM, the structural variations, polymorphism and heterogeneity within amyloid populations can be quantified. Furthermore, we present evidence that rare species in the heterogeneous fibril populations formed in vitro matches those observed in ex vivo human patient brain samples, giving support to the view that disease and physiologically relevant amyloid species can be formed in vitro. The approach we present links complimentary population level information to high-detailed structural analysis methods such as cryo-TEM and ssNMR which can be used to determine the average core structures of one or few dominant A$\beta_{42}$ polymorphs to atomic detail[36,47]. This opens up important questions regarding the uniqueness of the molecular surfaces of individual amyloid fibril polymorphs. It also opens up exciting possibilities for integrative structural analysis approaches that link information on population, structural variation and rare species obtained from AFM with cryo-TEM and ssNMR derived high resolution structural maps of few dominant species for obtaining a fuller understanding of diverse disease relevant and ex vivo amyloid population holistically. Small changes in assembly conditions applied here revealed that the rugged landscape of polymorphic amyloid assembly can easily be shifted, although our data demonstrate that a common set of polymorphs, including disease related polymorph structures, can exist across different conditions despite the sensitivity of the polymorph distribution. Therefore, understanding the precise in vitro conditions that generate in vivo disease associated structures in rarity or in dominance will be important for resolving the assembly mechanisms of these structures in a medical disease setting and providing valuable experimental tools for diagnostics and therapeutics research and development. Overall, the conclusions from this study link in vitro and in vivo structural data and put into focus the structural variations, diversity, heterogeneity and molecular population level properties of amyloid in general. Determining the factors that control the formation of dominant amyloid polymorph structures,

control the formation of rare species and control the extent of structural variations and population heterogeneity, are all required for a fuller understanding of the roles that amyloid molecular populations play in biology, as cause or consequence of disease or as structures required for functions.

## Methods

### Expression, purification and characterisation of monomeric Aβ$_{42}$

Recombinant Aβ$_{42}$ was expressed and purified based on a protocol by Linse, Walsh and co-workers[37,38]. Briefly, *E. coli* BL21 cells were transformed with the pET-sac Aβ$_{42}$ plasmid. Colonies were grown on and selected from LB-agar plates to form starter cultures in LB media. 1 L volumes of culture were then grown until they reached an OD$_{600}$ of 0.6 at which point protein expression was induced using IPTG and the cultures were left for 4 more hours before cell pellets were harvested by centrifugation. The protein was expressed in $4 \times 1$ L batches and purified 1 L at a time. Aβ$_{42}$ containing inclusion bodies were purified by sonicating frozen cell pellets in 20 mM Tris buffer at pH 8 followed by centrifugation at $18,000 \times g$. The inclusion bodies were further sonicated in Tris buffer to dissolve the pellet followed by immediate immersion in 8 M urea to induce denaturing conditions. The protein was purified by ion exchange chromatography using DEAE sepharose followed by SEC using a 60 ml Superdex 75 Increase column (GE healthcare) in 6 M GuHCl. Chromatograms and SDS-PAGE data (Supplementary Fig. S1a, b) provided evidence of purified monomeric Aβ$_{42}$. Finally, immediately before use, monomeric Aβ$_{42}$ was further purified using a 30 ml Superdex 75 Increase column (GE healthcare). Alternately, several aliquots of Aβ$_{42}$ were concentrated using centrifugal filtration units (Amicon) and then purified using the 30 ml Superdex 75 Increase column twice. Either way the sample was eluted into the appropriate buffer solution of 20 mM sodium phosphate (NaPh) pH 8.0, sodium phosphate pH 7.4, Tris pH 7.4 or HEPES pH 7.4 (at pH 8.0 followed by adjustment to pH 7.4 if necessary once eluted from the column). The ability of the expressed and purified Aβ$_{42}$ monomers to assemble and form β-rich and ThT positive amyloid fibrils was confirmed by CD and ThT kinetics assay, respectively (Supplementary Fig. S1c, d). To assess the cytotoxic potential of the Aβ$_{42}$ samples, monomeric protein solutions were diluted to a concentration of 50 µM in 10 mM HEPES buffer (10 mM HEPES, 50 mM NaCl, 1.6 mM KCl, 2 mM MgCl$_2$, 3.5 mM CaCl$_2$) and left at room temperature for 2–3 h to oligomerise as detailed in earlier publication[39]. These were added to cell culture media of mouse primary hippocampal neurons to a final concentration of 10 µM and the cytotoxicity effect of the samples on the cells was assayed after 1 week using ReadyProbes™ (Thermofisher). Cells were incubated for 15 mins with one drop each of ReadyProbes™ blue (all cells) and green (dead cells) and imaged using a Zeiss CO widefield microscope with DAPI and FITC filters, and the cytotoxic potential of the oligomeric species was shown to be comparable to a commercially produced Aβ$_{42}$ peptide purchased from rPeptide (Supplementary Fig. S1e).

### Analysis of published experimental conditions for amyloid assembly

A literature analysis was conducted where the most highly cited publications involving at least one experiment in which monomeric Aβ$_{42}$ was present and assembled in some capacity, from 2005 to 2020 were identified. The experimental conditions from each of the identified publications was tabulated (Supplementary Table S1). These conditions were analysed, the types of experimental buffers used were annotated and their relative frequencies in the publication dataset quantified. Specifically, buffers that were intended as experimental buffers for Aβ$_{42}$ polymerisation were recorded. For example, for the frequently employed strategy in which monomeric peptide was solubilised in NaOH or DMSO before being diluted into an experimental buffer or cell culture media, the particular buffer or media were included in the analysis not the initial step involving NaOH or DMSO. Another commonly used method is to use a 'vehicle' buffer in which monomer is initially incubated before being transferred to cell culture. In this case the 'vehicle' buffer *would* be recorded as it is the initial condition in which substantial polymerisation could occur. Included in the tabulated publication data but not the analysis are publications detailing Aβ$_{42}$ fibril structures that can be found in the PDB but did not meet the criteria for being one of the most highly cited in the year it was published. The literature search was performed initially using Web of Science, followed by validation by cross-referencing using Google Scholar.

### In vitro fibril formation

To prepare amyloid fibril samples formed from the recombinant Aβ$_{42}$, the monomer solution eluted from the final step of purification was immediately incubated at 37 °C without agitation. The solution was incubated for at least 1 week before being sonicated for 5 s. Freshly sonicated fibril fragments were subsequently used as seeds and were mixed with freshly prepared monomer to a final concentration of 15 µM monomer equivalent concentration with 1% total protein mass in pre-formed seed fibrils. These second generation fibrils were incubated at 37 °C without agitation for at least 1 week in the appropriate buffer solution before used for AFM imaging.

### AFM specimen preparation and imaging

Each sample to be used for AFM imaging was diluted to 10 µM using a combination of the appropriate buffer and a solution of HCl that had been pre-determined to result in a final pH of 2 when mixed in a 1:20 ratio with the appropriate buffer for each sample. Immediately after dilution, 20 µl samples were deposited onto freshly cleaved mica surfaces (Agar scientific, F7013) and incubated for 5 min. This low pH deposition method was employed to adjust the surface charge of the mica surfaces to allow more efficient deposition of the fibrils. Following the 5 min incubation, the sample was washed with 1 ml of filter sterilised milli-Q water and then dried using a gentle stream of nitrogen gas. To confirm that the brief pH jump deposition method did not perturb the fibrils, fibrils that were subjected to the procedure were used to generate seeds for a seeded reaction in which the thioflavin T profile was found to be identical to that from a reaction using seeds made from fibrils which had not undergone the brief pH jump procedure. Fibrils were imaged using a Multimode 8 AFM with a Nanoscope V controller (Bruker) operating under peak-force tapping mode using ScanAsyst probes (silicon nitride triangular tip with tip height = 2.5–2.8 µm, nominal tip radius = 2 nm, nominal spring constant 0.4 N/m, Bruker). Each collected image had a scan size of $4 \times 4$ µm and $2048 \times 2048$ pixels or $8 \times 8$ µm and $4096 \times 4096$ pixels. Therefore, the same pixel density is maintained for all images within the dataset. A scan rate of 0.305 Hz was used for the $4 \times 4$ µm and 0.2 Hz for the $8 \times 8$ µm scans. The instrument parameters noise threshold (related to the expected size-range of the features) and Z limit was set to 0.5 nm and 1.5 µm, respectively. When obtaining high-detail topographical maps (Fig. 2 and Supplementary Fig. S3), the fibrils were probed gently using ~400 pN of force, which is substantially lower than the reported force required for the deformation of amyloid fibrils[73]. These imaging settings coupled with the sample deposition conditions resulted in a low typical background noise in the image data, with an average standard deviation of the background pixel values of 0.255 nm. Nanoscope analysis software (Version 1.5, Bruker) were used to process the image data by flattening the height topology data to remove tilt and scanner bow.

### Individual filament structural analysis

Aβ$_{42}$ fibrils on the AFM height images were traced and digitally straightened[74] using algorithms in Trace_y software[34,45] and the height profile for each fibril was extracted from the centre contour line of the straightened fibrils. The periodicity of the fibrils was then determined using fast-Fourier transform of the height profile of each fibril. The final structural datasets consist of a varying multitude of images comprising of 100 individually traced fibrils per assembly condition used. In order to determine appropriate fibrils for analysis, selection criteria were applied in which single fibrils, with a free segment up to the point at which they either reached an end or overlapped with another fibril were used. Selected fibrils also had at least 3 repeating units and no visible breakages. This criterion also ruled out randomly bundled filaments which do not display periodic repeating

patterns along the filament. Care was also taken to ensure that no more than one segment of the same fibril was included in the analysis unless a distinct change in morphology was observed in which case the segments were treated as a separate fibrils. By excluding broken fibrils likely occurred upon deposition, it is possible that the polymorph distribution results were biased against thin filaments with a specific twist pattern.

Straightened fibril traces were corrected for the tip-sample convolution effect by CPR-AFM based on rigid-body geometric modelling of the tip-fibril contact points[34,45]. Briefly, the algorithm first corrects for the lateral dilation of nano-structures resulting from the finite dimensions of the AFM probe, without the loss of structural information, by resampling of the fibrils at tip-sample contact points. This results in recovering subpixel resolution of lateral sampling. Each pixel value in the straightened fibril data is then corrected in their x, y and z coordinates, resulting in a contact point cloud set. The 3D surface envelope model of each filament was subsequently constructed (Supplementary Fig. S4). This was achieved by least-squares spline fitting of the local cross-sectional envelope represented by a cubic spline curve in the x/z-plane at gridded y-coordinates of the point cloud data along the helical axis aligned to the y-axis. For the reconstruction of the 3D models, filament helical symmetry was estimated by building 3D models with various symmetries from the data, back calculating a dilated AFM image and comparing the angle of the fibril twist pattern with that of the straightened fibril trace in the simulated images and in the 2D Fourier transform of the simulated images. Then for construction of the 3D surface envelopes for each of the individual fibrils, the degree of twist per pixel along the y-axis was found by dividing 360° with the product of fibril periodicity and its symmetry number (e.g. 1, 2 or 3, etc.). The 3D models were made assuming a helical symmetry using a moving-window approach, in which a window, centred at a pixel $n$, contained the pixels $n - y$ to $n + y$ where $y$ is the axial length covered by 180° twist. The central pixel $n$ is not rotated while neighbouring pixels on both sides along the y-axis are rotated by a rotation angle, which is the product of the twist angle and the distance from n in pixels. Rotation angle values are negative in one direction from n and positive in the other direction, with the specific direction depending on the handedness of the fibril, determined by manual inspection of the straightened fibril image and its 2D Fourier transform image. To visualise the cross-sections of the individual filaments, the cross-section contact-point density maps (Supplementary Fig. S4) was constructed using a non-parametric bivariate Normal kernel density estimator (Eq. 1) with 'untwisted' point clouds of the individual filaments where the point clouds are aligned to the helical axes system twisting along the y-axis for each filament[45].

$$\hat{f}_{\mathbf{H}}(\mathbf{x}; \mathbf{H}) = \frac{1}{n}\sum_{i=1}^{n} K_{\mathbf{H}}(\mathbf{x} - \mathbf{x}_i); \quad K_{\mathbf{H}} = \frac{1}{2\pi}|\mathbf{H}|^{-1/2}e^{-\frac{1}{2}\mathbf{x}^{\mathbf{T}}\mathbf{H}^{-1}\mathbf{x}} \quad (1)$$

In Eq. (1), $\hat{f}_{\mathbf{H}}$ is the estimated contact-point density, $\mathbf{x}$ are the $x$, $z$ coordinates vectors. $K_{\mathbf{H}}$ is the normal kernel distribution, and $\mathbf{H}$ is the bandwidth parameter. For the bandwidth parameter, a scalar bandwidth parameter estimated with a non-parametric method[75] times the identity matrix is used. The contact points used for the contact-point density maps were selected by manual inspection of the digitally straightened image data with the general principle that the lines of pixels used were within one radius distance away from the central axis on the original image (thus having small magnitude of dilation caused by tip-sample convolution). The CPR-AFM and individual filament 3D reconstructions were carried out using the open-source Trace_y analysis software (available at https://github.com/wfxue/Trace_y)[45].

## Morphometric analysis of individual fibrils and polymorph distribution analysis

From each of 3D-reconstructed fibril models, several morphometric parameters[29,36] were calculated. The average fibril heights, $h$, were measured on the central ridge of each fibril. The mean cross-over distance, $cod$, were measured between the peaks on the centre fibril height profile. The twist handedness of the fibril, $hnd$, is defined as −1 for left-hand twisted fibrils

and +1 for right-hand twisted fibrils. The handedness of twist is determined for each filament by manual inspection of the straightened fibril image and its 2D Fourier transform image. The directional periodic frequency, $dpf$, is calculated as $dpf = hnd/$helical pitch. The filament mean AFM tip accessible cross-sectional area, $csa$, of the fibril is calculated by polar integration along the fibril axis of the reconstructed fibril 3D surface envelopes. The filament mean cross-sectional radius to the helical axis, $csr$, is calculated as the average distance from the filament's AFM tip accessible surface to its helical axis. The filament cross-sectional mean second polar moment of area, $csjz$, is calculated by calculating the mean second moment of area with regard to the x and y-axis of the cross-section using the perpendicular axis theorem.

## Agglomerative hierarchical clustering

For the agglomerative hierarchical clustering analysis, a $L_1$ (Manhattan) distance measure, $d_\xi$, was defined to measure the dissimilarity between a pair of fibrils $v$ and $w$. The $d_\xi$ distance score takes into account the differences in the helical properties of the fibrils (along the helical z-axis) characterised by the $dpf$ values, and the differences in the cross-sectional shape (in the x/y-cross-sectional plane) of the two fibrils Eq. (1).

$$d_\xi(v, w) = d_{cs}(v, w) + d_{dpf}(v, w) \quad (2)$$

In Eq. (2), the cross-sectional shape difference $d_{cs}$ is the standardised RMSD between the average tip-accessible fibril cross-sections defined according to Eq. (3).

$$d_{cs}(v, w) = \sqrt{\frac{1}{n_{cs}}\sum_{i=1}^{n_{cs}}\left(r_{v,i} - r_{w,i}\right)^2}/\sigma_{cs} \quad (3)$$

In Eq. (3), r are the surface envelope coordinates in helical-polar coordinate system where the x-y axis plane rotates around the helical z-axis with the same periodicity as the helical periodicity of the individual helical filament. The parameter $n_{cs}$ is the number of sampling points, in this case, 1° intervals were sampled for a total of 360 points used. The parameter $\sigma_{cs}$ is the standardisation parameter, with the standard deviation of $d_{cs}$ of the whole dataset used here. The distance, $d_{dpf}$, in Eq. (2) is defined in Eq. (4).

$$d_{dpf}(v, w) = \frac{\left|dpf_v - dpf_w\right|}{\sigma_{dpf}} \quad (4)$$

For the standardisation parameter $\sigma_{dpf}$ in Eq. (3), the standard deviation of $d_{dpf}$ of the whole dataset used. The heterogeneity index HI is calculated over $n_{d\xi}$ pairwise distances in a set of fibrils using Eq. (5) and its standard error estimated using the Jackkife method

$$HI = RMS_{d_\xi} = \sqrt{\frac{1}{n_{d_\xi}}\sum_{i=1}^{n_{d_\xi}} d_{\xi,i}^{\,2}} \quad (5)$$

Agglomerative hierarchical clustering was performed using the average distance linkage function shown in Eq. (6) for cluster $r$ and $s$.

$$d_\xi(r, s) = \frac{1}{n_r n_s}\sum_{i=1}^{n_r}\sum_{j=1}^{n_s} d_\xi(v_{ri}, w_{sj}) \quad (6)$$

Explained briefly, the shortest distance, $d_\xi$, between any two fibrils within a data set is found. Those two data points are then considered to be joined into a cluster and the two individual fibril data points are no longer used to determine the next cluster. Instead the average distance of both fibril data points to other fibrils or clusters are used. This is then repeated until all of the data is linked under one cluster.

## Comparative structural matching of individual fibril 3D-envelopes to cryo-TEM data

The pairwise similarity between each and every individually reconstructed 3D surface envelopes of the 400 $A\beta_{42}$ fibrils analysed and cryo-TEM derived structural maps of $A\beta_{42}$ and $A\beta_{40}$ listed in Table 1 were quantified. The structural maps were downloaded from the EMDB. The downloaded maps were axis aligned, and subsequently used to generate simulated AFM images (Fig. 5a) as well as to estimate their tip-accessible cross-sections calculated according to the method previously described[36] using an average tip radius of 5.5 nm. The pairwise $d_\xi$ distance scores in Eqs. (2)–(4) were subsequently calculated and ranked. The clusters identified to best match a cryo-TEM entry is found by finding individual fibrils with $d_\xi$ scores of less than 0.5 to the relevant cryo-TEM entry, and with other individual cluster members having $d_\xi$ scores of less than 1.0 to the same cryo-TEM entry and to each other within the cluster. To estimate the likelihood of random positive matches by chance, the structural matching analysis was also carried out with a randomly generated AFM fibril dataset where the fibril envelopes were generated using 4000 random combinations of the morphometric parameters of maximum cross-sectional radius, minimum cross-sectional radius, cross-sectional symmetry and helical twist and pitch length. The bounds of the morphometric parameters are those seen in the experimental data (Supplementary Table 2), but they are otherwise randomised in the simulation with the only rule that the minimal cross-sectional radius must be smaller than the maximal cross-sectional radius. Only 0.0036% of the pairwise distances were below $d_\xi$ scores of 0.5 (3 out of 84000 pairwise comparisons), confirming that the probability of obtaining a positive match by chance is very small compared to the observed frequency for the matched clusters of rare species.

## Dissociated neuronal cultures

Experimental and animal handling procedures were carried out in accordance with the UK—Animals (Scientific Procedures) Act 1986 and satisfied the local regulatory body at the University of Sussex. Primary hippocampal cultures were prepared from P0/P1 C57Bl6 mice. Briefly, pups were sacrificed via cervical dislocation and the hippocampus isolated and washed three times with pre-warmed/pre-buffered 1× Minimal Essential Media (MEM) (Gibco Life Technologies) containing 10% FCS, 3% glucose, 1% sodium pyruvate and 1% pen/strep (Gibco Life Technologies). Using a 1 ml pipette, the tissue was triturated until fully dissociated and diluted further with supplemented MEM. Neurons were seeded at a density of $6 \times 10^4$ cells/well on 12 mm glass coverslips (Fisher) pre-coated with poly-D-lysine $(20\,\mu g \cdot ml^{-1})$ and laminin $(20\,\mu g \cdot ml^{-1})$ (Sigma-Aldrich). 2–4 h later plating media was topped up with Neurobasal medium without phenol red (Gibco Life Technologies), supplemented with 2% B27, 1% Glutamax (Gibco Life Technologies) and 1% pen/strep. The final volume was 4× that of the original plating volume. After 4–5 days incubation, cells were treated with 3.25 μM cytosine arabinoside (Sigma-Aldrich) to halt astrocyte proliferation and half the media volume was exchanged to provide fresh nutrients and further dilute the original plating media. Cultures were kept in conditions of 5% CO2/95% air at 37 °C until experimentation at DIV14—21.

## Assay of synaptic function

To label functional presynaptic terminals, neurons were incubated for 60 s with 10 μM FM1-43FX in EBS (Extracellular bath solution, composition (mM)) supplemented with 20 μM AP-V and 50 μM CNQX. Cultures were stimulated with 600APs at 20 Hz to mobilise and fluorescently label all vesicles in the recycling pool. Following a second 60 s incubation to ensure completion of endocytosis, excess dye was washed from the membrane surface by exchanging EBS x7. Active synapses could be recognised easily by the presence of distinct puncta along neuronal processes. Neurons were fixed in 2% PFA and anti-synaptophysin 1 used to visualise all terminals regardless of activity. Images were acquired using a PicoXpress imager and quantified using ImageJ. Regions of interest were identified using the synaptophysin staining. An IsoData algorithm was applied to threshold the image, and this used to generate a mask isolating presynaptic terminals. Co-localisation between this mask and FM1-43FX labelled synapses was calculated using Mander's coefficient M1 and the proportion of active synapses determined.

## Reporting summary

Further information on research design is available in the Nature Portfolio Reporting Summary linked to this article.

## Data availability

All data on individual filaments and systematic literature study that support the findings of this study are available within the paper and its Supplementary Information (Supplementary Figs. S3, S4 and S5, Supplementary Tables S1, S2 and S3).

## Code availability

The code reported in this paper is publicly available as an open-source software at https://github.com/wfxue/Trace_y.

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

## Acknowledgements

We gratefully thank Sara Linse for the generous gift of $A\beta_{42}$ expression plasmid used in this study. We thank the members of the Xue group and the Serpell group for helpful comments throughout the preparation of this manuscript. We also thank Ian Brown for technical support. This work was supported by funding from the Biotechnology and Biological Sciences Research Council (BBSRC), UK grant BB/S003312/1 and BB/Z516880/1, BBSRC South Coast Biosciences Doctoral Training Partnership (SoCoBio DTP) grant BB/T008768/1 and Engineering and Physical Sciences Research Council (EPSRC), UK DTP grant EP/R513246/1.

## Author contributions

L.D.A. designed the research, conducted the experiments and analysed the data. L.L. wrote the analytical software tools and analysed the data. K.F. and T.J.P. conducted the experiments and analysed the data. N.L.W. analysed the data. L.C.S. designed the research and analysed the data. W.F.X. designed the research, wrote the analytical software tools, analysed the data and managed the research. The manuscript was initially drafted by L.D.A and W.F.X. and edited through contributions of all authors.

## Competing interests

The authors declare no competing interests.
