## [Transparent Peer Review file · Communications Chemistry]

Structural reconstruction of individual filaments in A β 42 fibril populations assembled in vitro reveal rare species that resemble ex vivo amyloid polymorphs from human brains

Corresponding Author: Dr Wei-Feng Xue

This manuscript has been previously reviewed at another journal. This document only contains information relating to versions considered at Communications Chemistry.

Point-to-point response to the Reviewers' comments

Reviewer #4:

We thank reviewer #4 for the comments raised and for donating their valuable time to read our manuscript.

The point remains that AFM does not read internal registry and interfaces directly. The argument provided by the authors still rests on modeling and correlation, not an independent internal readout.

This reviewer also commented in a previous revision.

1. The surface-envelope matching obtained by AFM is necessary but not sufficient for the conclusion. Cryo-EM polymorphs are determined by the internal β -sheet registry, backbone conformation and protofilament interfaces. These features were not resolved by AFM. Two distinct polymorphs could yield similar surface envelopes, so convergent morphology cannot be excluded.

Reply: Once again, this reviewer does not appear to believe that the AFM based contact-point densities we obtained (for example those in the previous revised manuscript Fig 5b-e) showing the 3D surface envelopes of amyloid polymorphs are sufficient to identify the specific polymorphs. This is an opinion not supported by the large amounts of evidence obtained by us as well as other researchers in the community, and in the large amounts of cryo-EM based structural data available in the EMDB and the PDB, which show that the surface envelopes are particularly sensitive fingerprints that can be used for polymorph identification. Thus, full 'internal readout' is not necessarily needed for identification purposes since the tightly packed fibril cores lead to highly unique molecular surface information.

In the previous version of the manuscript, we have already clarified in the result section that "only the surface envelopes of the individually analysed fibrils were reconstructed and matched", and we were very careful in our wording to say that the evidence suggests "that the synthetic fibrils formed *in vitro* from recombinant A β ₄₂ can represent, in rare parts, disease relevant fibril structures formed *in vivo*". We have also already highlighted precedents of successful AFM-based molecular identifications of amyloid polymorphs that were confirmed by cryo-EM, validating the scientific validity and usefulness of our approach (e.g. references 36 46 and 47).

Contrary to reviewer #4's comment, our approach does not rely on finding statistical correlations. We do carry out modelling of the molecular surfaces based on cryo-EM derived maps. However, this is based directly on the map data and the level of modelling required is no more complicated conceptually than the modelling required in cryo-EM or crystallographic studied to fit a molecular model into the observed densities.

To facilitate further community discussions on the uniqueness of the molecular surfaces of the amyloid cores, we have now modified the text in the discussion section (page 21-22 in

the revised manuscript) to include “This opens up important questions regarding the uniqueness of the molecular surfaces of individual amyloid fibril polymorphs. It also opens up exciting possibilities for integrative structural analysis approaches that link information on population, structural variation and rare species obtained from AFM with cryo-TEM and ssNMR derived high resolution structural maps of few dominant species for obtaining a fuller understanding of diverse disease relevant and ex vivo amyloid population holistically”.

2. The new data support the overall pathogenic relevance of the populations (toxicity and synaptic impairment), but they do not demonstrate seeding or toxicity for the rare matched polymorphs specifically, nor do they present seeding assays. My request was stricter (strain-like behavior of the matched species), and remains unaddressed experimentally.

Reply: The specific biological contributions of the rare matching fibril species in our amyloid populations cannot currently be determined separately from the whole population. This is because the rare species only represent a few percentages of the whole population, and the fact that prior to the advances presented in this manuscript, the polymorph distribution could not be resolved to sufficient detail. Thus, the advances we presented in this manuscript includes the prerequisite technology needed for any experimental work on characterising (e.g. the seeding, toxicity and strain-like behaviour), enriching or targeting specific, potentially rare, amyloid polymorphs in the future. We have now added on page 18 of the revised manuscript the following: “the approach presented here to resolve the amyloid polymorph distributions will unlock future experimental opportunities on characterising (e.g. the seeding, toxicity and strain-like behaviour), enriching or targeting specific, potentially rare, amyloid polymorph species” to discuss potential future experiments.